# TGF-β-Induced PAUF Plays a Pivotal Role in the Migration and Invasion of Human Pancreatic Ductal Adenocarcinoma Cell Line Panc-1

**DOI:** 10.3390/ijms252111420

**Published:** 2024-10-24

**Authors:** Miso Lee, Hyejun Ham, Jiyeong Lee, Eun Soo Lee, Choon Hee Chung, Deok-Hoon Kong, Jeong-Ran Park, Dong-Keon Lee

**Affiliations:** 1Division of Research Program, Scripps Korea Antibody Institute, Chuncheon 24341, Gangwon-do, Republic of Korea; smile0203@skai.or.kr (M.L.); hhj@skai.or.kr (H.H.); codory03@skai.or.kr (J.L.); kong0131@skai.or.kr (D.-H.K.); 2Department of Internal Medicine, Research Institute of Metabolism and Inflammation, Yonsei University Wonju College of Medicine, Wonju 26426, Gangwon-do, Republic of Korea; es1121@hanmail.net (E.S.L.); cchung@yonsei.ac.kr (C.H.C.)

**Keywords:** PAUF, pancreatic ductal adenocarcinoma, Epithelial–mesenchymal transition, TGF-β, Smads, MEK-ERK signaling pathway

## Abstract

Pancreatic adenocarcinoma upregulated factor (PAUF) was initially identified as a secreted protein that is substantially expressed in pancreatic ductal adenocarcinoma (PDAC). PAUF also affects invasiveness, motility, and the proliferation of cells in several types of cancer. Recently, PAUF was reported to play a pivotal role in the TLR4-mediated migration and invasion of PDAC cells. However, the mechanism inducing PAUF expression and its functional role in TGF-β-stimulated PDAC cells have not yet been studied. Thus, we first assessed whether TGF-β regulates PAUF expression in several PDAC cell lines and found a significant increase in PAUF expression in Smad signaling-positive Panc-1 cells treated with TGF-β. We also found that the PAUF promoter region contains a Smad-binding element. TGF-β-treated Panc-1 cells showed an increase in PAUF promoter activity, but this effect was not observed in TGF-β-stimulated Smad4-null BxPC-3 cells. Restoring Smad4 expression increased the PAUF promoter activity and expression in Smad4-overexpressing BxPC-3 cells treated with TGF-β. We further found that PAUF aggravated the TGF-β-induced epithelial–mesenchymal transition (EMT) in Panc-1 and BxPC-3 cells via the activation of MEK-ERK signaling. These results indicate that TGF-β/Smad signaling-mediated upregulation of PAUF plays a crucial role in EMT progression by activating the TGF-β-mediated MEK-ERK signaling pathway.

## 1. Introduction

Pancreatic ductal adenocarcinoma (PDAC) is the most prevalent neoplastic disease of the pancreas, accounting for more than 90% of all pancreatic malignancies [1], and has a five-year overall survival rate of approximately 10%. The high mortality caused by PDAC is particularly attributed to its high metastatic capacity [2].

Epithelial–mesenchymal transition (EMT) is a crucial event in cancer metastasis that leads to the migration and invasion of cancer cells. It is characterized by downregulation of epithelial markers, including E-cadherin, claudin, ZO-1, and cytokeratin, which are essential in mediating epithelial cell junctions (cell–cell adhesion) and cell integrity, and upregulation of mesenchymal markers, including vimentin, fibronectin, and N-cadherin, which have been linked to increased cell motility and the invasive phenotype [3,4]. Transforming growth factor-beta (TGF-β) has been identified as a potent inducer of EMT during carcinogenesis [5,6].

TGF-β is a secreted pleiotropic cytokine that plays crucial roles in immunoregulation, wound healing, angiogenesis, and cancer. The “canonical” TGF-β signaling pathway is mediated by the activation of Smad transcription factors. The binding of TGF-β to its type 2 receptor (TGF-βR2) elicits the formation of TGF-βR2 and TGF-βR1 heterodimer complexes, leading to phosphorylation of the Gly-Ser (GS) domain of TGF-βR1 [7], and the subsequent phosphorylation of Smad2 and Smad3. The phosphorylated Smads interact with cytoplasmic Smad4, and the Smad2/3/4 complex then translocates to the nucleus to regulate the transcription of target genes [8,9]. The canonical pathway is involved in the tumor-suppressive effects in cancer cells by the transcriptional upregulating of the cell cycle arrest-related (p21) and proapoptotic (Bim) genes [10]. In addition, TGF-β also activates “non-canonical” pathways, including the mitogen-activated protein kinase (JNK, p38 MAPK, ERK1/2) [11] and phosphatidylinositol 3-kinase (PI3K)/Akt pathways [12], independently of Smad proteins, and its signaling modulators are associated with the promotion and aggressiveness of tumors. In particular, the MEK-ERK signaling pathway is a pivotal driver of TGF-β-mediated EMT progression in cancer [13] and is essential for the migration and invasion of various PDAC cells [14]. Collectively, TGF-β-mediated Smad and non-Smad signaling pathways play paradoxical roles in the growth and metastasis of cancer.

Approximately 50–55% of PDAC cases exhibit functional loss of TGF-β/Smad signaling, involving a Smad4 deficiency, for example, via homozygous deletion and intragenic mutation. This functional loss is involved in carcinogenesis, invasion, and metastasis by activating TGF-β-mediated MEK1/2-ERK1/2 signaling pathways (Smad4-independent pathways) [15]. Consistently, a previous study reported that MEK1/2 and ERK1/2 phosphorylation was increased in Smad4-null BxPC-3 cells stimulated with TGF-β [16]. Accordingly, TGF-β plays a central role in the EMT-mediated progression of PDAC by upregulating the MEK1/2 and ERK1/2 signaling cascades.

Pancreatic adenocarcinoma upregulated factor (PAUF; also known as ZG16B) is a novel tumor-promoting secreted protein that plays a critical role in metastasis and cancer progression in several types of cancer, such as pancreatic cancer [17], cervical cancer [18], colorectal cancer [19], and ovarian cancer [20]. In pancreatic cancer, PAUF promotes tumorigenesis by acting as a tumor microenvironment (TME) regulator in an autocrine/paracrine manner [21]. Recently, the binding of PAUF with toll-like receptor 4 (TLR4) was reported to elicit the TLR4-mediated migration and invasion of PDAC cells [22]. These findings indicate that PAUF plays an important role in promoting carcinogenesis and metastasis via activation of the TLR4 signaling pathway. However, the regulatory mechanism underlying the expression of PAUF and its biological functions have not yet been studied in TGF-β-stimulated PDAC cells.

In the present study, we found that TGF-β/Smad signaling-induced PAUF promotes EMT progression by activating the MEK-ERK signaling cascade, leading to increased cell migration and invasion.

## 2. Results

### 2.1. Evaluation of TGF-β-Induced PAUF Expression in Pancreatic Ductal Adenocarcinoma (PDAC) Cell Lines

We first investigated whether TGF-β regulates the expression of PAUF in different PDAC cell lines (CFPAC-1, BxPC-3, Panc-1, MIA PaCa-2, and AsPC-1). Among the five cell lines tested, Panc-1 cells exhibited significant increases in both mRNA and protein levels of PAUF following stimulation with TGF-β, compared to the PBS (Veh) control (Figure 1A,B). Next, we evaluated the expression of PAUF in TGF-β-stimulated Panc-1 cells at different time points. Treatment with TGF-β resulted in significant increases in the mRNA and protein levels of PAUF in Panc-1 cells at 8 h of treatment, compared with that in the Veh, and these increases were enhanced until 24 h (Figure 1C,D). TGF-β promotes EMT, triggering changes in the cell shape, loss of cell polarity, and reorganization of cytoskeleton, and leads to tumor invasion and metastasis [23]. To assess the correlation between TGF-β-mediated EMT and PAUF expression, Panc-1 cells were treated with TGF-β for 24 h. The stimulated cells showed EMT-mediated spindle-shaped morphology and increases in intra- and extracellular PAUF expression (Figure 1E,F). These results indicate that TGF-β induces PAUF expression in Panc-1 cells.

### 2.2. TGF-β-Mediated PAUF Upregulation Requires Smad2/3 Phosphorylation and Smad4 Expression

Smad molecules, such as R-Smads (Smad2, Smad3) and Co-Smad (Smad4), are central mediators of the canonical TGF-β pathway [24] and play critical roles in effectuating TGF-β-mediated EMT, leading to tumor progression [25]. However, a few PDAC cell lines exhibit dysregulation of TGF-β/Smad signaling via genetic loss of Smad4 or Smad2/3, inactivating mutation [26,27,28,29]. To evaluate the correlation between TGF-β/Smad signaling and PAUF expression, PDAC cell lines were stimulated with TGF-β. We first confirmed the levels of phosphorylated Smad2/3 and Smad4 expression in Smad4-intact (Panc-1 and MIA PaCa-2) and Smad4-deficient (CFPAC-1, AsPC-1, and BxPC-3) PDAC cell lines [30,31]. As expected, the basal expression of Smad4 was confirmed in both untreated and TGF-β-treated Panc-1 and MIA PaCa-2 cells but not in CFPAC-1, AsPC-1, and BxPC-3 cells. TGF-β-treated BxPC-3 and Panc-1 cells, but not CFPAC-1, MIA PaCa-2, and AsPC-1 cells, showed increased Smad2/3 phosphorylation (Figure 2A). These data indicate that Panc-1 is a TGF-β/Smad signaling-positive PDAC cell line. TGF-β-induced PAUF expression may be regulated through TGF-β/Smad signaling in TGF-β/Smad-positive PDAC cell lines (Figure 1). Therefore, we next explored the signal mediators involved in TGF-β-mediated PAUF expression in Panc-1 cells using a TGF-β receptor I kinase inhibitor (SB-431532). TGF-β treatment significantly increased the phosphorylation of Smad2/3, and the phosphorylation was effectively inhibited by SB-431532, without affecting Smad4 expression (Figure 2B). As expected, this inhibitor abrogated the TGF-β-induced increase in mRNA and protein levels of PAUF (Figure 2C). These effects were confirmed by immunofluorescence staining (Figure 2D). Additionally, we examined the effects of Smad (Smad2/3 and Smad4) knockdown on PAUF expression in TGF-β-treated Panc-1 cells. Smad-knockdown Panc-1 cell lines were successfully generated and validated using immunoblot analysis (Figure 2E). As expected, knockdown of Smad2/3 or Smad4 alone or the knockdown of both markedly abolished the TGF-β-induced increase in mRNA, protein, and secreted levels of PAUF compared with that in cells transfected with siRNA control (Ctrl) (Figure 2F,G). Next, we performed a gain-of-function experiment restoring Smad4 expression in BxPC-3 cells using a Smad4 expression vector (pcSmad4). Ectopic expression of Smad4 was confirmed using immunoblot analysis (Figure 2H). TGF-β treatment did not affect the mRNA and protein levels of PAUF in BxPC-3 cells. However, following the restoration of Smad4 expression, TGF-β stimulation drastically increased the expression of PAUF (Figure 2I,J). Taken together, these data support the possibility of the TGF-β-mediated activation of Smad signaling transcriptionally inducing PAUF expression in TGF-β/Smad-positive PDAC cells.

### 2.3. Smad-Binding Element (SBE) of the PAUF Promoter Is Crucial for Transcriptional Upregulation of the PAUF Gene

To assess whether TGF-β-activated Smads directly regulate PAUF transcription, we first analyzed the transcription factor-binding motifs in the PAUF promoter region (1.7 kb) using the PROMO program 3.0.2 http://alggen.lsi.upc.es, accessed on 19 April 2024) and identified putative SBE (5′-AGAC-3′) at –804/–801 bp (Figure 3A). Smad3 and Smad4, but not Smad2, also interact directly with SBE (5′-AGAC-3′ or 5′-GTCT-3′) in the promoter regions of target genes. We next examined which Smads are involved in TGF-β-mediated transcriptional upregulation of PAUF in Panc-1 cells using a TGF-β receptor I kinase inhibitor (SB-431532). Treatment with TGF-β increased the binding of pSmad3 to SBE within the PAUF promoter region and the PAUF promoter activity, and these increases were blocked by SB-431532 (Figure 3B,C). To investigate whether the knockdown of Smads regulates a TGF-β-induced increase in the PAUF transcriptional activity, Panc-1 cells were transiently transfected with the control (Ctrl), Smad2/3, or Smad4 siRNAs alone or in combination and stimulated with the vehicle or TGF-β. TGF-β-induced binding of pSmad3 to the SBE of PAUF promoter was effectively inhibited by Smad2/3 single- and Smad2/3 plus Smad4 double-knockdown; however, the effects of TGF-β were not ameliorated by the knockdown of Smad4 alone (Figure 3D). These results indicated that genetic loss of Smad4 did not affect the binding of pSmad3 to SBE. As expected, the Smad2/3 single- and Smad2/3 plus Smad4 double-knockdown abolished the increase in TGF-β-mediated PAUF promoter activity. Interestingly, the interaction of pSmad3 with the SBE of PAUF promoter in Smad4-knockdown Panc-1 cells stimulated with TGF-β did not affect the increase in PAUF promoter activity (Figure 3E). Consistent with our findings, some previous studies reported that the MH1 domain of Smad3 plays a pivotal role in SBE binding, without interacting with other Smads, such as Samd2 and Smad4 [32]. In addition, Smad4 is not required for the nuclear translocation of Smad2 and Smad3, but is needed for the activation of transcriptional responses [33]. We further examined whether restoring Smad4 expression would transcriptionally upregulate the PAUF gene in Smad4-null PDAC cell line BxPC-3 stimulated with TGF-β. Treatment with TGF-β significantly promoted the binding of pSmad3 to the SBE of PAUF promoter in mock-transfected BxPC-3 cells; however, this binding did not affect the increase in transcriptional activity of the PAUF promoter, whereas restoration of Smad4 expression effectively increased the PAUF promoter activity in pcSmad4-transfected BxPC-3 cells (Figure 3F,G). Furthermore, TGF-β critically increased the transcriptional activity of the REDD1 promoter (SBE^*wt*^), but not of its SBE site mutants (SBE^*Mut*^ and SBE^∆*mut*^) in Smad4-intact Panc-1 cells, and this increase was not observed in mock-transfected BxPC-3 cells treated with TGF-β. However, treatment with TGF-β markedly recovered the transcriptional activity of the PAUF promoter (SBE^*wt*^), but not of its SBE site mutants in pcSmad4-transfected BxPC-3 cells (Figure 3H–J). These results indicate that Smads indeed interact with the SBE site of the PAUF promoter. Collectively, these findings strongly support the notion that PAUF is upregulated by TGF-β via the activation of Smad signaling.

### 2.4. PAUF Knockdown Inhibits TGF-β-Induced EMT in Panc-1 Cells

To explore whether TGF-β-induced PAUF expression is involved in effectuating EMT, we first examined the effects of PAUF knockdown in TGF-β/Smad signaling-positive Panc-1 cells stimulated with TGF-β. The PAUF knockdown cells were successfully generated and validated using immunoblotting and a sandwich enzyme-linked immunosorbent assay (ELISA) (Figure 4A,B). The Smad-independent MEK-ERK signaling pathway plays a central role in TGF-β-mediated EMT in cancer [13,34] and is required for the migration and invasion of PDAC cells [14]. Therefore, we examined the effects of PAUF knockdown on TGF-β-activated MEK-ERK signaling in Panc-1 cells. PAUF knockdown significantly decreased TGF-β-induced increases in MEK1/2 and ERK1/2 phosphorylation, whereas transfection with control siRNA had no effects (Figure 4C). Because TGF-β treatment upregulates EMT-inducing transcription factors, such as SNAI1 and ZEB1, via activation of MEK-ERK signaling, it upregulates the expression of mesenchymal markers, such as N-cadherin, vimentin, and α-SMA, and downregulates the expression of epithelial markers, such as E-cadherin, claudins, and ZO-1, which are involved in tumor invasion, metastasis, and cell motility [4,35]. Therefore, we assessed the regulatory effect of PAUF on the expression of TGF-β-mediated EMT-related modulators in Panc-1 cells. PAUF knockdown drastically suppressed the TGF-β-induced upregulation of SNAI1, ZEB1, vimentin, and α-SMA, whereas it recovered the downregulation of epithelial markers, such as E-cadherin and ZO-1, compared with that in control siRNA-transfected cells (Figure 4D). These results were confirmed using E-cadherin- or α-SMA-targeted immunofluorescence staining (Figure 4E). Under the same experimental conditions, knockdown of PAUF markedly decreased the TGF-β-induced increases in morphological changes, migration, and invasion, whereas transfection with control siRNA had no effect (Figure 4F). These findings suggest that TGF-β-induced PAUF plays an important role in elevating the EMT via activation of the MEK-ERK signaling pathway.

### 2.5. PAUF Overexpression Augments TGF-β-Induced EMT in Panc-1 and BxPC-3 Cells

We further investigated whether PAUF overexpression regulates the progression of TGF-β-mediated EMT in Smad signaling-positive Panc-1 cells. A PAUF-overexpressing Panc1 cell line was successfully generated and validated using immunoblotting and a sandwich ELISA (Figure 5A,B). We then examined the effects of PAUF overexpression on EMT-associated MEK-ERK signaling in Panc-1 cells treated with TGF-β. Ectopic expression of PAUF significantly augmented TGF-β-mediated increases in MEK1/2 and ERK1/2 phosphorylation. In particular, PAUF overexpression also increased the phosphorylation of MEK1/2 and ERK1/2, albeit to a lesser extent, in Panc-1 cells not stimulated with TGF-β (Appendix A). In addition, ectopic expression of PAUF increased the relative expression of MEK-ERK signaling-regulated EMT-inducing transcription factors, such as SNAI1 and ZEB1, in TGF-β-stimulated or unstimulated Panc-1 cells, compared with that in cells transfected with the control vector (Lenti-Ctrl). Moreover, PAUF overexpression also markedly augmented the TGF-β-induced increase in mRNA and protein levels of mesenchymal markers, such as α-SMA (encoded by ACTA2) and vimentin (encoded by VIM), whereas it further reduced the TGF-β-mediated decrease in the expression of epithelial markers, such as E-cadherin (encoded by CDH1) and ZO-1 (encoded by TJP1), compared with that in control vector-transfected Panc-1 cells (Figure 5C,D). As expected, ectopic expression of PAUF further increased morphological changes, migration, and invasion in untreated or TGF-β-treated Panc-1 cells, compared with those in control vector-transfected cells (Figure 5E). To assess the involvement of atypical expression of PAUF on EMT progression, we further examined whether the ectopic expression of PAUF regulates the TGF-β-mediated EMT progression in Smad4-null BxPC-3 cells. A PAUF-overexpressing BxPC-3 cell line was successfully generated and validated using immunoblotting, immunofluorescence staining, and a sandwich ELISA. Notably, TGF-β treatment did not affect the expression of PAUF in control- or PAUF-overexpressing BxPC-3 cells, indicating a requirement for the intact function of Smads on the TGF-β-mediated increase in PAUF expression (Figure 5F–H). Next, the regulatory effect of PAUF on the activation of TGF-β-mediated MEK-ERK signaling was investigated in BxPC-3 cells. PAUF overexpression augmented MEK1/2 and ERK1/2 phosphorylation in BxPC-3 cells with or without TGF-β stimulation, compared with that in control vector-transfected cells (Appendix A). Moreover, its overexpression further increased the TGF-β-induced expression of SNAI1, ZEB1, vimentin, and α-SMA, whereas it diminished the TGF-β-mediated decrease in E-cadherin and ZO-1 levels via activation of MEK-ERK signaling to a greater extent, compared with that in control vector-transfected cells (Figure 5I). Consistent with these findings, the PAUF-overexpressing BxPC-3 cells showed EMT-related spindle-shaped morphology and increases in cell motility, such as migration and invasion, without affecting TGF-β treatment, and these biological processes were prominently augmented by TGF-β, compared with those in control cells (Figure 5J). Collectively, these results support the possibility that PAUF plays a pivotal role in increasing EMT via activation of the TGF-β-mediated MEK-ERK signaling pathway.

## 3. Discussion

TLRs are mainly expressed on immune cells and participate in the regulation of innate and adaptive immunity. They are also highly expressed on various cancer cells, promoting the inflammatory TME to induce the proliferation, survival, invasion, and metastasis of cancer cells [36]. Of the many subtypes of TLRs, TLR4 is highly expressed on several types of PDAC cells, and its expression correlates with the invasiveness of cancer cells [37,38]. Recently, direct PAUF-TLR4 binding was reported to evoke TLR4-mediated migration and invasion in TLR4 high-expressing PDAC cells, such as AsPC-1 and BxPC-3 cells [22].

EMT is a critical event in cancer progression, which elicits cell invasion and metastasis. By activating the Smad-independent pathway, TGF-β acts as a pivotal inducer of EMT. However, the regulatory mechanism of TGF-β-mediated PAUF expression and its biological functions have not been reported in PDAC cells.

The TGF-β/Smad signaling pathway exerts multiple biological functions, including differentiation, extracellular matrix association, cell-cycle progression, motility, and death, through the transcriptional regulation of various Smad-dependent target genes [39]. In the present study, we first found that TGF-β markedly increases PAUF expression in Smad signaling-positive Panc-1 cells [30,31], which indicates that PAUF expression might be regulated by the activation of TGF-β/Smad signaling; however, similar results were not observed in Smad signaling-defective PDAC cell lines, such as MIA PaCa-2, CFPAC-1, AsPC-1, and BxPC-3 cells [30,31]. We therefore investigated TGF-β/Smad signaling modulators that affect the PAUF expression induced by stimulating TGF-β using a TGF-β receptor I kinase inhibitor (SB-431532) or Smad-targeting siRNAs in Panc-1 cells. The treatment with SB-431532 or transfection with Smad-targeting siRNAs abolished the TGF-β-induced increase in the mRNA, protein, and secreted levels of PAUF. Additionally, the recovery of PAUF expression was observed upon restoring Smad4 in Smad4-null BxPC-3 cells. Our data indicate that the TGF-β-mediated upregulation of Smad signaling transcriptionally induces the expression of PAUF in Smad signaling-positive PDAC cells.

Smad proteins exert their transcriptional regulatory activities by interacting with specific DNA-binding motifs (SBEs) [8]. Smad3 and Smad4, but not Smad2, are able to bind to the SBE sequences [32,40,41,42]. TGF-β stimulation was reported to result in the phosphorylated-Smad2/3 heterodimer complex translocating to the nucleus in a Smad4-independent manner. TGF-β-treated Smad4-null BxPC-3 cells showed an increase in the nuclear translocation of the p-Smad2/3 complex, but it was not sufficient to activate reporters for TGF-β-induced transcriptional responses, which were restored by ectopic expression of wild-type Smad4 [33]. These data suggest that Smad4 is not required for the nuclear translocation of p-Smad2/3 but is essentially for the TGF-β-induced transcriptional activation of Smad-dependent target genes. In this study, we first identified that the PAUF promoter region contains a putative SBE motif in the –804 to –801 region. Therefore, we sought to determine whether TGF-β-activated Smads directly regulate transcriptional activity of the PAUF promoter. Based on the experimental data obtained through ChIP-PCR and the Luc-reporter assay of the PAUF promoter (SBE^*WT*^, SBE^*Mut*^, and SBE^∆*mut*^), we concluded that TGF-β-mediated activation of Smads plays a pivotal role in the transcriptional upregulation of the PAUF gene.

The Ras-Raf-MEK-ERK signaling pathway regulates cellular processes, such as differentiation, proliferation, and cell survival, but its abnormal activation leads to tumorigenesis and metastasis via EMT promotion [43]. TGF-β increases EMT by activating the Smad-independent Ras-Raf-MEK-ERK-AP-1 signaling cascade, which upregulates the transcription of E-cadherin repressor gene, SNAI1. As a transcription repressor, SNAI1 is involved in the transcriptional downregulation of epithelial markers, such as E-cadherin (encoded by CDH1) [44]. The SNAG domain of SNAI1 binds to the CDH1 promoter [45] and recruits histone deacetylases (HDACs). Subsequently, SNAI1, mSin3A, HDAC1, and HDAC2 form a multi-repressor complex that suppresses the expression of E-cadherin [46]. ZEB1 is also an E-cadherin-transcriptional repressor induced during EMT. SNAI1 elicits the nuclear translocation of Ets1, and its translocation is required for ZEB1 expression [47]. The recruitment of CtBP and transcriptional corepressors (HDAC1 and HDAC2) following the direct binding of ZEB1 to the CDH1 promoter leads to the transcriptional downregulation of the CDH1 gene, resulting in inhibition of E-cadherin expression and induction of EMT [48]. Consequently, suppression of E-cadherin correlates with the upregulation of mesenchymal markers, such as Fibronectin, vimentin, and α-SMA [49]. Similarly, our data indicate that TGF-β treatment induced the upregulation of transcription repressors, such as SNAI1 and ZEB, by activating the MEK-ERK signaling cascade, leading to the downregulation of epithelial markers, such as E-cadherin and ZO-1, whereas it upregulated the expression of mesenchymal markers, such as vimentin and α-SMA. Moreover, we found that PAUF knockdown prevented EMT-related cell migration and invasion via the suppression of TGF-β-mediated MEK-ERK signaling. In contrast, ectopic expression of PAUF showed increases in migration and invasion by activating the MEK-ERK signaling cascade without stimulating TGF-β, and these processes were further enhanced by TGF-β treatment in Panc-1 and BxPC-3 cells. Thus, our results suggest that PAUF might play a crucial role in TGF-β-mediated EMT progression by activating the MEK-ERK signaling cascade.

PAUF is crucial for carcinogenesis and metastasis as it activates TLR4-mediated MEK-ERK signaling [50,51]. Moreover, PAUF directly binds to TLR4 and is involved in promoting EMT in TLR4 high-expressing PDAC cells, such as BxPC-3 and AsPC-1. However, MIA PaCa-2, Capen-1, and Panc-1 cells are TLR4 very low-expressing PDAC cell lines. [22]. Our data clearly indicate that ectopic expression of PAUF significantly enhances cell migration and invasion through the activation of MEK-ERK signaling, independent of TGF-β, in both Panc-1 and BxPC-3 cells. Additionally, these processes are further promoted by TGF-β. Accordingly, PAUF may activate not only TLR4 but also other receptors that regulate the MEK-ERK signaling pathway related to EMT.

Abnormal activation of receptor tyrosine kinases (RTKs), such as EGFR, VEGFR2/3, and PDGFR-β, is also involved in the dysregulation of MEK-ERK signaling, leading to tumor progression [52,53,54,55,56]. However, to date, a direct correlation between these receptors and TGF-β-induced PAUF has not been reported. Further studies should be conducted to verify the exact regulatory mechanisms of EMT progression through interaction between TGF-β-induced PAUF and RTKs (e.g., EGFR, VEGFR2/3, and PDGFR-β).

Our data indicate that TGF-β stimulation induces the expression of PAUF via the activation of the Smad-dependent signaling pathway. However, we did not investigate the involvement of non-Smad signaling, such as JNK, p38 MAPK, and ERK1/2 in TGF-β-mediated PAUF expression. Thus, we could not exclude the possibility that non-Smad signaling-mediated transcription factors are involved in TGF-β-mediated expression of PAUF. Additional studies are necessary to confirm the mechanism of induction of PAUF expression via TGF-β-mediated non-Smad signaling.

Collectively, our findings show, for the first time, that TGF-β/Smad signaling-induced PAUF plays pivotal roles in cell migration and invasion via activating the MEK-ERK signaling cascade (Figure 6).

## 4. Materials and Methods

### 4.1. Reagents and Chemicals

The antibody for PAUF (cat# MAB7777) was purchased from R&D systems, Inc. (Minneapolis, MN, USA). Antibodies for α-smooth muscle actin (α-SMA; cat# sc-53142), vimentin (cat# sc-373717), ZO-1 (cat# sc-33725), SNAI1 (cat# sc-271977), ZEB1 (cat# sc-515797), SMAD4 (cat# sc-7966), and β-actin (cat# sc-47778) were obtained from Santa Cruz Biotechnology (Dallas, TX, USA). Antibodies against p-ERK1/2 (cat# 9101S), ERK1/2 (cat# 9102S), p-MEK1/2 (cat# 9121S), MEK1/2 (cat# 9122S), p-SMAD2 (cat# 3108S), pSMAD3 (cat# 9520S), SMAD2/3 (cat# 8685S), and E-cadherin (cat# # 3195S) were purchased from Cell Signaling Technology (Beverly, MA, USA). Recombinant human TGF-β and TGF-β receptor I kinase inhibitor, SB-431542, were purchased from R&D systems and Sigma-Aldrich (St. Louis, MO, USA), respectively. siRNAs targeting human Smad2/3 (cat# sc-37238), Smad4 (cat# sc-29484), and ZG16B/PAUF (cat# sc-93479) were purchased from Santa Cruz Biotechnology, and pCMV6-XL5 (cat# PCMV6XL5), pCMV-Smad4 (cat# SC116771), pLenti-control (cat# PS100092), and pLenti-PAUF (cat# RC202244L3) vectors were obtained from OriGene Technologies, Inc. (Rockville, MD, USA).

### 4.2. Cell Culture

Human PDAC cell lines CFPAC-1, BxPC-3, Panc-1, MIA PaCa-2, and AsPC-1 were obtained from the American Type Culture Collection (ATCC; Manassas, VA, USA). The CFPAC-1 cell line was cultured in Iscove’s modified Dulbecco’s medium (IMDM; ATCC). BxPC-3 and AsPC-1 cell lines were maintained in RPMI-1640 medium (Corning Inc., New York, NY, USA) and Panc-1 and MIA PaCa-2 cell lines were cultured in Dulbecco’s minimal essential medium (DMEM; Gibco, Palo Alto, CA, USA). All media were supplemented with 10% fetal bovine serum (FBS; Gibco) and 1% penicillin/streptomycin (Gibco) and the cells were incubated in a 5% CO_2_ humidified incubator.

### 4.3. Reverse Transcription PCR (RT-PCR) and Quantitative Real-Time PCR (qRT-PCR)

Total RNA was isolated from cultured cells using the TRIzol reagent (Invitrogen, Carlsbad, CA, USA). cDNA was synthesized from 1 μg of total RNA using M-MLV Reverse Transcriptase (Promega, Madison, WI, USA), according to the manufacturer’s protocols. mRNA levels of PAUF and GAPDH were determined using RT-PCR with target gene-specific primers. The primer sequences of the genes used for analysis are listed in Table 1. To quantitatively evaluate the mRNA levels of target genes, 0.5 μg of total RNA was converted to cDNA using ReverTra Ace^TM^ qPCR RT Master Mix with gDNA Remover (Toyobo, Osaka, Japan), following the manufacturer’s protocols. qRT-PCR analysis was performed using Power SYBR^TM^ Green PCR Master Mix (Thermo Fisher Scientific, Waltham, MA, USA) on a real-time PCR cycler QuantStudio3 (Thermo Fisher Scientific) using target gene-specific primers. The sequences of the primers are shown in Table 2. The relative mRNA levels were calculated using the 2^−∆∆CT^ method, employing the housekeeping gene *GAPDH* as an internal control. All assays were performed in triplicate in three independent experiments.

### 4.4. Immunoblotting

Protein was extracted by lysing the cells in RIPA lysis buffer (Sigma-Aldrich). The protein concentration in cell lysate was measured using the BCA assay. The lysates were separated on 8–15% sodium dodecyl sulfate-polyacrylamide gel and transferred onto polyvinylidene fluoride membranes using a transfer device. The blots were blocked with 3% bovine serum albumin (BSA) or 5% skim milk in Tris-buffered saline (T&I, Seoul, Korea) containing 0.1% Tween-20 (Sigma-Aldrich) for 1 h and immunoblotted using the indicated antibodies, as described previously [57]. The information regarding the antibodies is presented in Section 4.1. Protein bands were quantitated densitometrically using the ImageJ software 1.54i (accessed on 3 March 2024) (National Institutes of Health, Bethesda, MD, USA). The densitometric evaluation of protein bands was analyzed using GraphPad Prism 6.0 (GraphPad Software Inc., San Diego, CA, USA).

### 4.5. Immunocytochemistry

To assess whether TGF-β induces PAUF expression in Panc-1 cells, TGF-β-treated cells were cultured on glass slides in 12-well plates for 24 h until they reached 60–70% confluence. The treated cells were fixed with 3.7% formaldehyde and permeabilized with 0.1% saponin solution. After the slides were rinsed twice with phosphate-buffered saline (PBS), they were blocked with 3% BSA for 2 h and incubated overnight with mouse monoclonal anti-PAUF (R&D systems; 1:500) at 4 °C. The slides were then washed with PBS and labeled with Alexa Fluor 488 Goat anti-mouse IgG (Thermo Fisher Scientific; 1:1000) at 4 °C for 2 h. To evaluate the inhibitory effects of SB-431542 (Sigma-Aldrich) on Smad-mediated PAUF expression in Panc-1 cells treated with TGF-β, the cells were stimulated with or without TGF-β (10 ng/mL) for 24 h after pretreated with SB-431542 (10 μM) for 1 h. The fixed, permeabilized, and blocked slides of the cells were incubated overnight with mouse monoclonal anti-PAUF (1:500) at 4 °C. The slides were rinsed with PBS and labeled with Alexa Fluor 488 Goat anti-mouse IgG (1:1000) at 4 °C for 2 h. To investigate the effects of PAUF knockdown on TGF-β-induced EMT in Panc-1 cells, the cells were transfected with 100 nM scrambled- or PAUF-targeting siRNA, and grown to 60–70% confluence on glass slides in 12-well plates. The transfected cells were then treated with or without TGF-β for 24 h, fixed with 3.7% formaldehyde, and permeabilized with 0.1% saponin solution. The slides were rinsed twice with PBS and blocked with 3% BSA for 2 h. The slides were then incubated overnight with rabbit polyclonal anti-E-cadherin (Cell Signaling Technology; 1:500) or mouse monoclonal anti-αSMA (Santa Cruz Biotechnology; 1:500) at 4 °C, washed with PBS, and labeled with Alexa Fluor 488 Goat anti-rabbit IgG (Thermo Fisher Scientific; 1:1000) or Alexa Fluor 647 Goat anti-mouse IgG (Thermo Fisher Scientific; 1:1000) at 4 °C for 2 h. To confirm the successful generation of a BxPC-3 PAUF-overexpressing stable cell line, the fixed, permeabilized, and blocked slides of the cells were incubated overnight with mouse monoclonal anti-PAUF (1:500) at 4 °C. The slides were washed with PBS and labeled with Alexa Fluor 488 Goat anti-mouse IgG (1:1000) at 4 °C for 2 h. For nuclei staining, the slides were further incubated with 1 µg/mL 4′,6-diamidino-2-phenylindole (DAPI; Sigma-Aldrich) for 15 min at room temperature. All the slides were mounted with Faramount Aqueous Mounting Medium (Dako, Glostrup, Denmark), and images of slides were observed using a Zeiss LSM710 confocal microscope (Carl ZEISS, Berlin, Germany) at 400× magnification. The ImageJ software was used for analyzing the fluorescence intensity.

### 4.6. Construction of the Promoter Vector and Luciferase Assay

The PAUF promoter (1.7 kb) was amplified from genomic DNA isolated from Panc-1 cells using PCR with specific primers (forward: 5′-TACTCGAGAGGTGGTTGGCTGG-3′; reverse: 5′-ATAAGCTTTGCCGGGCACCCTC-3′). The amplified PAUF promoter (1.7 kb) was cloned into the XhoI and HindIII restriction sites in the pGL3-basic vector. This recombinant plasmid DNA was named pLuc-PAUF (SBE*^WT^*). The mutant construct of the putative SBE site (AGAC, −804 to −801) in pLuc-PAUF (SBE*^WT^*) was generated via site-directed mutagenesis using the QuikChange site-directed mutagenesis kit (Agilent technologies, Santa Clara, CA, USA) and mutation-specific primers (forward: 5′-GAGGGGGAAAGGAGTGCTGAAGAAAACCAGCAGGGC-3′; reverse: 5′-GCCCTGCTGGTTTTCTTCAGCACTCCTTTCCCCCTC-3′), according to the manufacturer’s protocol. This mutant plasmid was named pLuc-PAUF (SBE*^Mut^*). For deletion mutagenesis of the putative SBE site, overlap extension PCR was performed using two-pair primers (P1-forward: 5′-TACTCGAGAGGTGGTTGGCTGG-3′; reverse: 5′-GTTTTCTCACTCCTTTCCCCCT-3′; P2-forward: 5′-AGAAAACCAGCAGGGCCCCG-3′; and reverse: 5′-ATAAGCTTTGCCGGGCACCCTC-3′), as described previously [58]. The SBE deletion construct of the PAUF promoter was cloned into the XhoI and HindIII sites in the pGL3-basic vector. This deletion mutant plasmid was named pLuc-PAUF (SBE^∆*mu*^*^t^*). Panc-1 or BxPC-3 cells (4 × 10^5^) were cotransfected with 100 nM Smad-targeting siRNAs or 1 μg of Smad4 expression vector (pCMV-Smad4; OriGene Technologies, Inc.) along with 1 μg of PAUF promoter-Luc vectors (SBE*^WT^*, SBE*^Mut^*, and SBE^∆*mu*^*^t^*) and 1 μg of the β-gal expression vector (pCMV-LacZ; Clontech, Palo Alto, CA, USA) using the Lipofectamine 3000 reagent (Invitrogen). At 6 h after transfection, the medium was replaced with complete growth medium and the cells were allowed to stabilize for 24 h, followed by treatment with or without TGF-β (10 ng/mL) in the presence or absence of SB-431542 (10 μM) for 24 h. The luciferase activity was assayed using the Luciferase Reporter Assay System (Promega), according to the manufacturer’s instructions, and the β-galactosidase activity was measured using ortho-nitrophenyl-β-galactoside (Sigma-Aldrich). The relative luciferase units were normalized against the β-galactosidase activity.

### 4.7. Chromatin Immunoprecipitation Assay

Panc-1 cells were treated with or without TGF-β (10 ng/mL) for 1 h after pretreatment with SB-431542 (10 μM) or transfection with 100 nM Smad-targeting siRNAs for 1 or 24 h. The ChIP assay was carried out using a Chromatin Immunoprecipitation (ChIP) Assay Kit (Millipore Corporation, Billerica, MA, USA), following the manufacturer’s instructions. A DNA/protein cross-linking complex was obtained by incubating the cells for 20 min at 37 °C in 1% formaldehyde. After sonication, p-Smad3/DNA complexes were immunoprecipitated with a rabbit polyclonal pSmad3 antibody (Cell Signaling Technology), and normal anti-rabbit IgG (Cell Signaling Technology) was used as a negative control. After DNA extraction, targeted promoter sequences of PAUF were identified via PCR using specific primers (forward: 5′-ACGAGCTTTTCTCAGCTGGCG-3′; reverse: 5′-GGAACTGGCGCTTCTCCAG-3′) spanning PAUF promoter regions containing the pSmad3 binding to the SBE (AGAC) sequence. The products (156 bp) were confirmed using gel electrophoresis.

### 4.8. PAUF Enzyme-Linked Immunosorbent Assay

To analyze the levels of secreted PAUF, Panc-1, Smad-knockdown Panc-1, PAUF-overexpressing Panc-1, or BxPC-3 cells were cultured in a DMEM or RPMI-1640 medium containing the vehicle or TGF-β (10 ng/mL) for the indicated times. The culture supernatants were concentrated using Amicon Ultra-15 Centrifugal Filter Units (Millipore) and collected to measure the levels of secreted PAUF. Plates were coated with anti-PAUF (5 µg/mL) for 24 h at room temperature and incubated with the collected supernatants at 37 °C for 2 h. Biotin-conjugated PAUF detection antibodies (250 ng/mL) were added, the plates were kept for 90 min at 37 °C, and subsequently incubated with streptavidin-HRP (1:5000) for 30 min at 37 °C. The levels of secreted PAUF were determined by measuring the absorbance at 450 nm using the BioTek Epoch 2 microplate spectrophotometer. These experimental procedures were performed as described previously [20,22].

### 4.9. Cell Migration and Invasion Assays

Cell migration and invasion assays were examined using 24-well Transwell^®^ plates with a Transwell insert with a 8 µm pore size membrane (Corning Inc.). For invasion assays, the upper chamber of the Transwell insert was coated with Matrigel^®^ Matrix (Corning), whereas this coating was not done for cell migration assays. The lower chamber of the Transwell was filled with 800 µL of culture medium containing 10% FBS. PAUF-knockdown Panc-1 (7 × 10^4^), PAUF-overexpressing-Panc-1 (7 × 10^4^), or -BxPC-3 (4 × 10^4^) cells were suspended in 200 µL of serum-free culture medium containing the vehicle or TGF-β (10 ng/mL), seeded into the upper chambers, and incubated for 24 h at 37 °C. For both assays, the cells were fixed with absolute methanol and stained with 0.1% crystal violet solution. The non-migrated cells on the inside of the upper chamber were completely removed using cotton swabs. The images of migrated cells were observed using an inverted microscope and the number of cells was counted in four fields per sample using the Image J software (National Institutes of Health).

### 4.10. Statistical Analyses

The GraphPad Prism 6.0 version software (GraphPad Software Inc.) was used for statistical analyses. The number of replicates is listed in the respective figure legends and the representative results were obtained from at least three independent experiments. All quantitative data are presented as mean ± standard deviation (S.D.). The statistical significance was determined using unpaired two-tailed *t*-tests for analysis between two groups or one-way ANOVA for single variable analysis or two-way ANOVA for two independent variable analysis followed by post-hoc multiple comparisons test with Bonferroni correction, depending on the experimental groups analyzed. A *p*-value < 0.05 was considered statistically significant.

## 5. Conclusions

We demonstrated that TGF-β/Smad signaling-induced PAUF promotes EMT through the activation of the MEK-ERK signaling pathway and plays a crucial role in the invasion and metastasis of pancreatic cancer (Figure 6). Our findings highlight a novel therapeutic strategy for inhibiting cancer progression, including the invasiveness and metastasis of Smad signaling-positive PDAC cells.

## Figures and Tables

**Figure 1 ijms-25-11420-f001:**
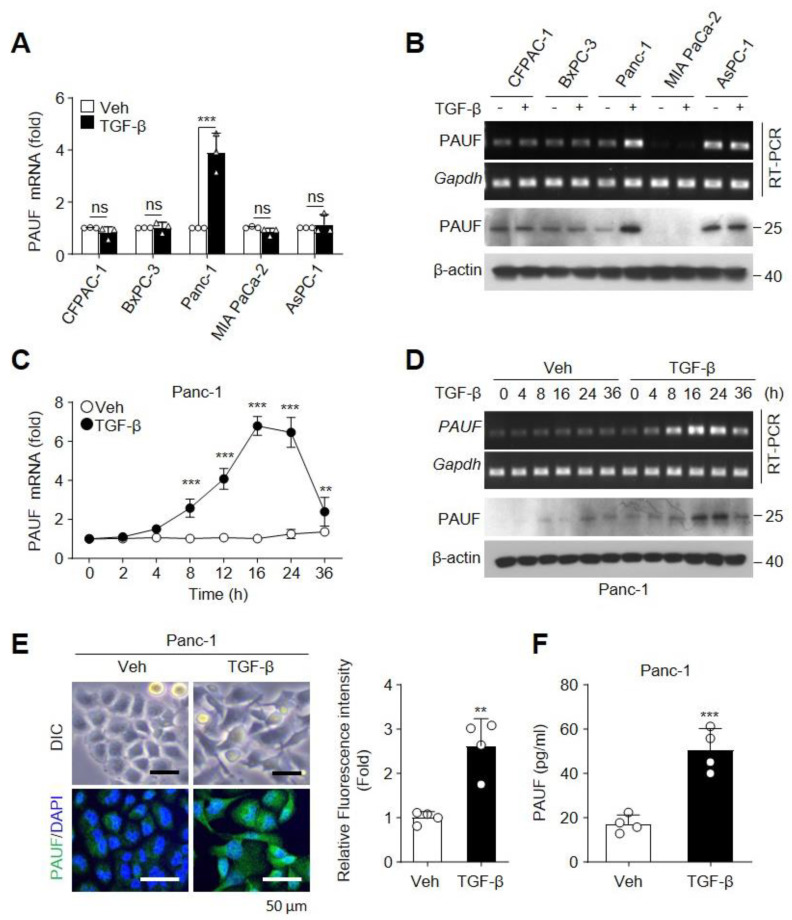
Expression of pancreatic adenocarcinoma upregulated factor (PAUF) is upregulated by TGF-β in Panc-1 pancreatic ductal adenocarcinoma (PDAC) cells. (**A**) Indicated PDAC cell lines were stimulated with vehicle or TGF-β (10 ng/mL) for 12 h. mRNA levels of PAUF were quantified using qRT-PCR (*n* = 3). (**B**) PDAC cell lines were treated with vehicle or TGF-β and analyzed at 24 h post-treatment. Levels of PAUF mRNA and protein were determined using RT-PCR and immunoblotting, respectively. Panc-1 cells were stimulated with vehicle or TGF-β for the indicated time periods. PAUF expression was analyzed using (**C**) qRT-PCR (*n* = 3), (**D**) RT-PCR, and immunoblotting. Panc-1 cells were treated with vehicle or TGF-β. After 24 h, (**E**) the cell morphology was observed using an optical microscope, and PAUF expression was measured via immunofluorescence staining with an Alexa 488-labeled anti-PAUF antibody; nuclei were stained with DAPI. Scale bar, 50 μm. The relative fluorescence intensity was measured in four fields per sample using the ImageJ software 1.54i (accessed on 3 March 2024) (*n* = 4). (**F**) ELISA for assessing extracellular PAUF levels in culture supernatants (*n* = 4). Statistical significance was assessed using (**A**,**C**) two-way analysis of variance (ANOVA) and (**E**,**F**) unpaired two-tailed *t*-tests. Data are presented as the mean ± standard deviation (SD). ns, not statistically significant, ** *p* < 0.01, *** *p* < 0.001.

**Figure 2 ijms-25-11420-f002:**
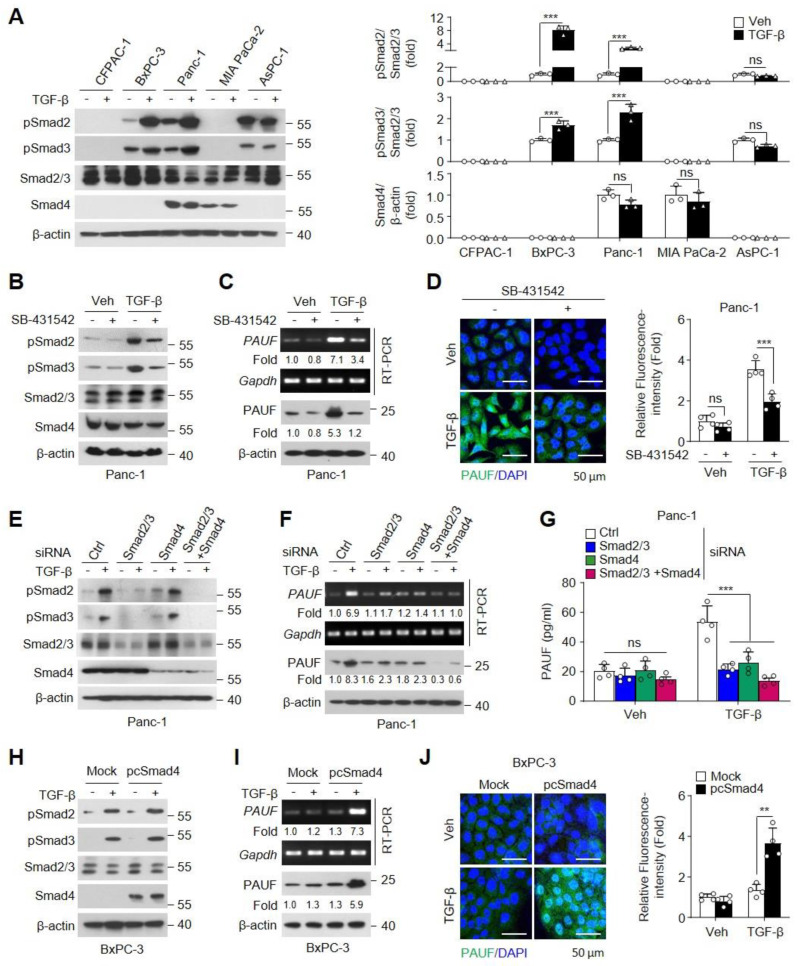
TGF-β-induced expression of pancreatic adenocarcinoma upregulated factor (PAUF) is mediated by activating the Smad signaling pathway. (**A**) Different PDAC cell lines were treated with vehicle or TGF-β (10 ng/mL) for 1 h. Phosphorylation and protein levels of TGF-β-mediated Smad signaling molecules were examined using immunoblotting. Densitometry was performed using the ImageJ software (*n* = 3). Panc-1 cells were pretreated with SB-431542 (10 μM) for 1 h and stimulated with vehicle or TGF-β for (**B**) 1 or (**C**,**D**) 24 h. (**B**) Phosphorylation and protein levels of Smads were determined using immunoblotting. (**C**) mRNA and protein levels of PAUF were determined using RT-PCR and immunoblotting, respectively. (**D**) Intracellular PAUF expression was confirmed through immunofluorescence staining with an Alexa 488-labeled anti-PAUF antibody and nuclei were stained with DAPI. Scale bar, 50 μm. Relative PAUF-expressing cells were quantitated in four fields per sample using the ImageJ software (*n* = 4). Panc-1 cells were transfected with 100 nM scrambled (Ctrl), Smad2/3, or Smad4 siRNAs alone or in combination, followed by treatment with vehicle or TGF-β for (**E**) 1 or (**F**,**G**) 24 h. (**E**) Phosphorylation and expression of Smad proteins were measured using immunoblotting. (**F**) mRNA and protein levels of PAUF were determined using RT-PCR and immunoblotting, respectively, and (**G**) secreted PAUF levels were measured in culture supernatants using ELISA (*n* = 4). BxPC-3 cells were transfected with control vector pCMV6 (Mock) or Smad4 expression vector pCMV-Smad4 (pcSmad4), followed by treatment with vehicle or TGF-β (10 ng/mL) for (**H**) 1 or (**I**,**J**) 24 h. (**H**) Smad2/3 phosphorylation and Smad4 expression were determined using immunoblotting. (**I**) PAUF expression was assessed using RT-PCR and immunoblotting. (**J**) Intracellular PAUF levels were visualized using immunofluorescence staining with an Alexa 488-labeled anti-PAUF antibody and nuclei were stained with DAPI. Scale bar, 50 μm. Relative PAUF-expressing cells were quantitated in four fields per sample in a randomized manner using the ImageJ software (*n* = 4). Statistical significance was calculated using two-way ANOVA followed by post-hoc multiple comparisons test with Bonferroni correction. Data are presented as the mean ± standard deviation (SD). ns, not statistically significant, ** *p* < 0.01, *** *p* < 0.001.

**Figure 3 ijms-25-11420-f003:**
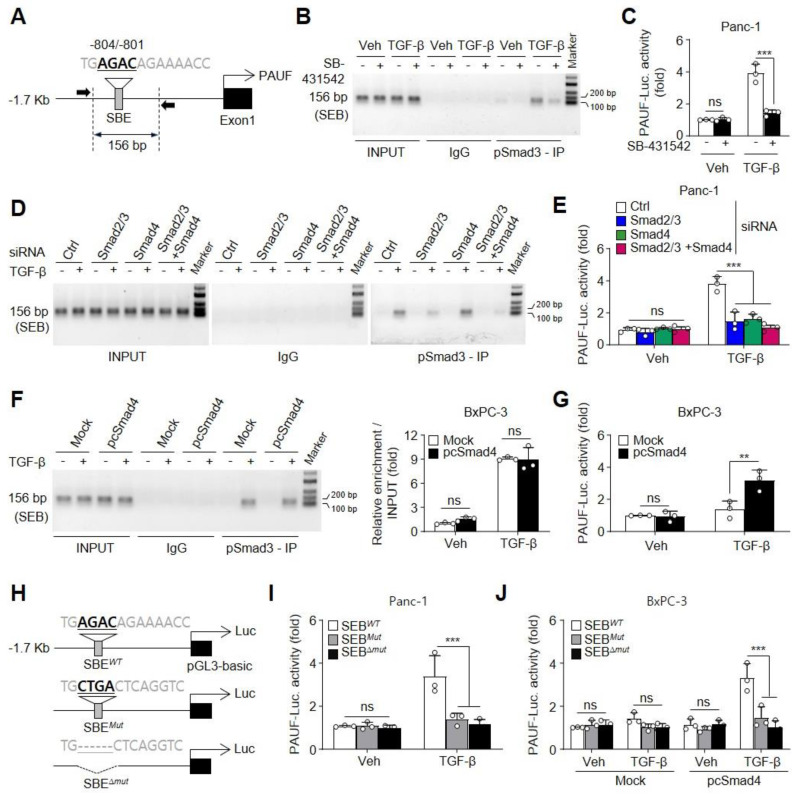
Smad-binding element (SBE) is essential for TGF-β-inducibility of the pancreatic adenocarcinoma upregulated factor (PAUF) promoter. (**A**) A schematic representation of the PAUF promoter (−1.7 Kb), highlighting the putative SBE used to analyze the PAUF promoter activity. (**B**) Panc-1 cells were pretreated with SB-431542 (10 μM) for 1 h and subsequently with vehicle or TGF-β (10 ng/mL) for 1 h. Binding of pSamd3 to SBE within the PAUF promoter was determined using the chromatin immunoprecipitation (ChIP) assay. (**C**) Panc-1 cells transfected with the PAUF promoter-Luc vector (pLuc-PAUF) were stimulated with vehicle or TGF-β for 24 h after pretreatment with SB-431542 (10 μM) for 1 h. The promoter activity was determined in cell lysates with a luminometer (*n* = 3). (**D**) Panc-1 cells were transfected with scrambled-, Smad2/3-, or Smad4-targeting siRNAs alone or in combination, followed by stimulation with or without TGF-β for 1 h to assess the binding of pSmad3 to SBE within the PAUF promoter using the ChIP analysis. (**E**) Panc-1 cells were transfected with the indicated siRNA alone or in combination with the pLuc-PAUF vector, and then treated with vehicle or TGF-β for 24 h to evaluate the PAUF promoter activity in cell lysates with a luminometer (*n* = 3). (**F**) BxPC-3 cells transfected with pCMV6 (Mock) or pCMV-Smad4 (pcSmad4) were stimulated with TGF-β for 24 h. Binding activity of pSmad3 to SBE within the PAUF promoter was analyzed using the ChIP assay. The fold intensity of ChIP enrichment for the SBE motif at −804/−801 was quantified and normalized against the input (*n* = 3). (**G**) BxPC-3 cells were transfected with mock or pcSmad4 in combination with the PAUF promoter-Luc vector, and then treated with vehicle or TGF-β for 24 h. The PAUF promoter activity was measured in cell lysates (*n* = 3). (**H**) Comparison of the putative SBE motif within the PAUF promoter (−1.7 Kb) with the SBE mutants. (**I**) Wild-type (SBE*^WT^*) and SBE site mutants (SBE*^Mut^* and SBE^∆*mut*^) of PAUF promoter-Luc vectors were transiently transfected into Panc-1 cells. After 24 h of transfection, the cells were stimulated with vehicle or TGF-β for 24 h and promoter activities were measured in cell lysates (*n* = 3). (**J**) BxPC-3 cells were transfected with mock or pcSmad4 in combination with PAUF promoter-Luc vectors (SBE*^WT^*, SBE*^Mut^*, or SBE^∆*mut*^), and then treated with vehicle or TGF-β for 24 h. The promoter activity was determined in cell lysates (*n* = 3). Statistical significance was determined using two-way ANOVA followed by post-hoc multiple comparisons test with Bonferroni correction. Data are displayed as the mean ± standard deviation (SD). ns, not statistically significant, ** *p* < 0.01, *** *p* < 0.001.

**Figure 4 ijms-25-11420-f004:**
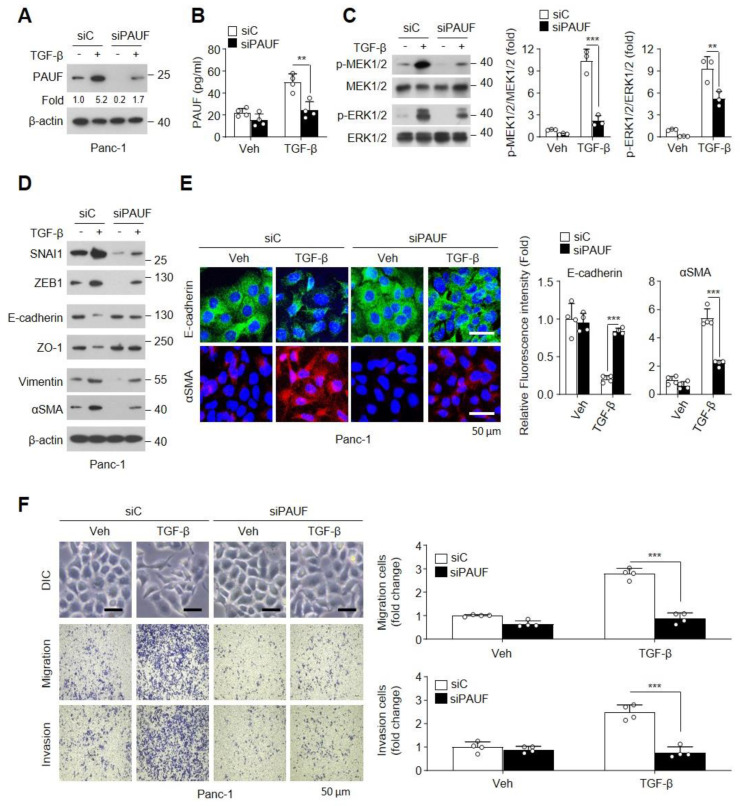
Depletion of pancreatic adenocarcinoma upregulated factor (PAUF) inhibits TGF-β-mediated increase in epithelial–mesenchymal transition (EMT) in Panc-1 cells. Panc-1 cells were transfected with scrambled (Ctrl) or PAUF siRNA and stimulated with or without TGF-β for (**C**) 1 or (**A**,**B**,**D**–**F**) 24 h. (**A**,**B**) Successful knockdown of PAUF expression was verified using immunoblotting and sandwich ELISA (*n* = 4). (**C**) Phosphorylation levels of MEK1/2 and ERK1/2 were examined using immunoblot analysis. Densitometry of the phosphorylated proteins was performed using the ImageJ software (*n* = 3). (**D**) Protein levels of EMT-related signaling molecules were determined using immunoblotting. (**E**) Expression levels of E-cadherin and αSMA were assessed using immunofluorescence staining with Alexa Flour 488-labeled anti-E-cadherin or Alexa Flour 647-labeled anti-αSMA antibody and nuclei were stained with DAPI. Scale bar, 50 μm. Relative fluorescence intensity was measured in four fields per sample in a randomized manner using the ImageJ software (*n* = 4). (**F**) Changes in cell morphology were observed using an optical microscope, and cell migration and invasion were evaluated using Transwell migration and Matrigel invasion assays. Scale bar, 50 μm. The migrated and invading cells were counted in four fields per sample (*n* = 4). Statistical significance was calculated using two-way ANOVA followed by post-hoc multiple comparisons test with Bonferroni correction. Data are represented as the mean ± standard deviation (SD). ** *p* < 0.01, *** *p* < 0.001.

**Figure 5 ijms-25-11420-f005:**
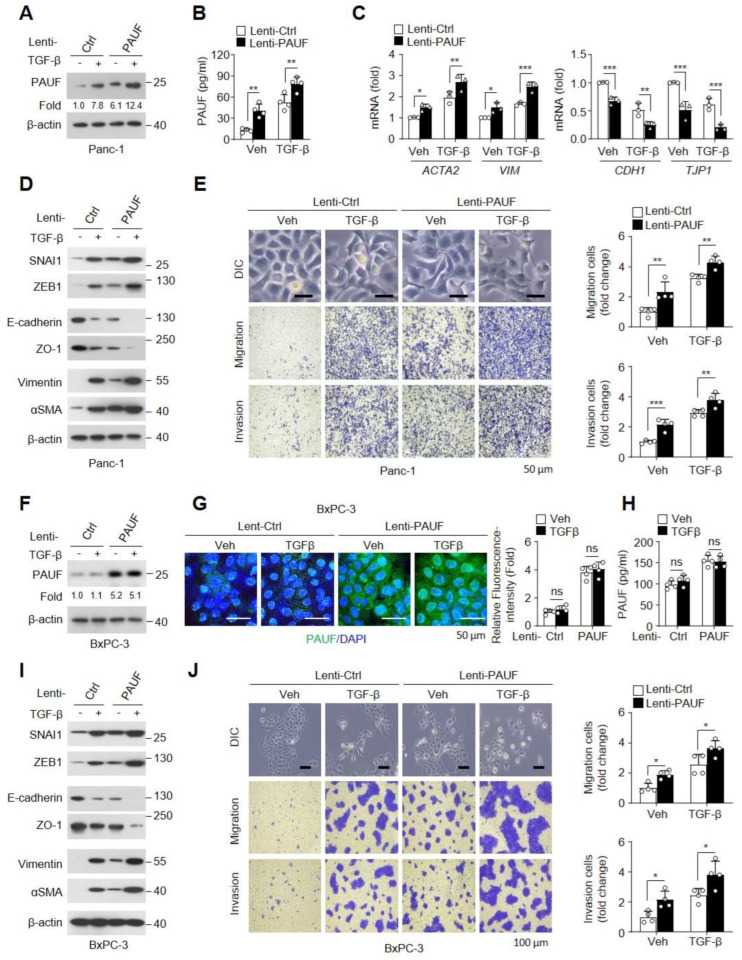
Overexpression of pancreatic adenocarcinoma upregulated factor (PAUF) aggravates TGF-β-increased cell motility and epithelial–mesenchymal transition (EMT)-like characteristics in Panc-1 and BxPC-3 cells. (**A**–**E**) Panc-1 cells were stably transfected with Lent-Ctrl or Lenti-PAUF expression vectors, and then stimulated with vehicle or TGF-β (10 ng/mL) for 24 h. (**A**,**B**) Successful ectopic expression of PAUF was confirmed using immunoblotting and sandwich ELISA (*n* = 4). (**C**) Relative mRNA levels of E-cadherin (*CDH1*), ZO-1 (*TJP1*), αSMA (*ACTA2*), and vimentin (*VIM*) were determined using qRT-PCR (*n* = 3). (**D**) Expression levels of EMT-related signaling proteins were analyzed using immunoblotting. (**E**) DIC images were used to observe the changes in cell morphology, and Transwell migration and Matrigel invasion assays were performed for measuring the EMT-mediated cell motility. Scale bar, 50 μm. The relative density of cell migration and invasion were measured in four fields per sample (*n* = 4). (**F**–**J**) Lenti-Ctrl- or Lenti-PAUF-transfected BxPC-3 cells were treated with or without TGF-β (10 ng/mL) for 24 h. Successful overexpression of PAUF was verified using (**F**) immunoblotting, (**G**) immunofluorescence staining, and (**H**) sandwich ELISA. For the immunofluorescence staining, the same procedures were conducted as described in Figure 2J. (**I**) Protein levels of EMT-related signaling molecules were determined using immunoblot analysis. (**J**) Changes in cell morphology were confirmed using an optical microscope, and cell migration and invasion were examined using Transwell migration and Matrigel invasion assays. Scale bar, 100 μm. Densitometry analysis of migration and invasion were measured in four fields per sample (*n* = 4). Statistical significance was evaluated using two-way ANOVA followed by post-hoc multiple comparisons test with Bonferroni correction. Data are displayed as the mean ± standard deviation (SD). ns, not statistically significant, * *p* < 0.05, ** *p* < 0.01, *** *p* < 0.001.

**Figure 6 ijms-25-11420-f006:**
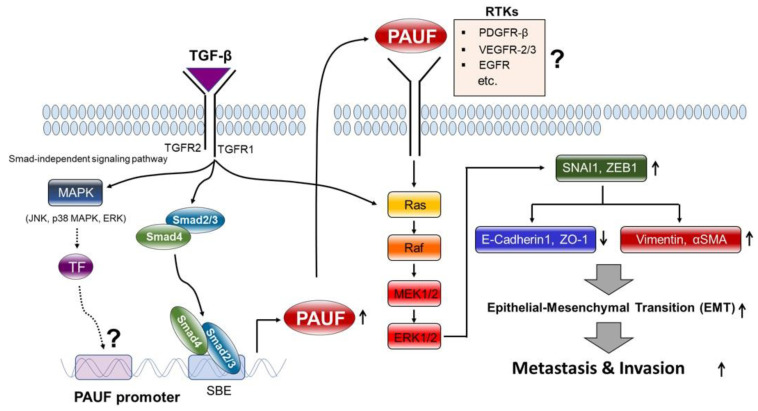
A schematic diagram illustrating the regulatory mechanism and functional roles of pancreatic adenocarcinoma upregulated factor (PAUF) in TGF-β-mediated epithelial–mesenchymal transition (EMT) signaling. The proposed mechanism involves a TGF-β/Smad signaling-induced increase in PAUF expression, which elicits the activation of MEK-ERK signaling cascade, upregulating transcription repressors (SNAI1 and ZEB1), leading to downregulation of epithelial markers (E-cadherin and ZO-1) and concomitant upregulation of mesenchymal markers (vimentin and αSMA), and resulting in the induction of EMT progression in Smad signaling-positive PDAC cells. The solid arrow represents the findings of our study, whereas the dashed arrow or a question mark indicates sections requiring further investigation. For details, please see the text.

**Table 1 ijms-25-11420-t001:** Sequences of gene-specific primers used in RT-PCR.

Gene	Forward Primer (5′–3′)	Reverse Primer (5′–3′)
*PAUF*	CATGAAATCACAGGGCTGCG	ATGTATTCGCCTGGCTGCA
*GAPDH*	ACATGTTCCAATATGATTCCACCC	ATGGACTGTGGTCATGAGTCCTT

**Table 2 ijms-25-11420-t002:** Sequences of gene-specific primers used in qRT-PCR.

Gene	Forward Primer (5′–3′)	Reverse Primer (5′–3′)
*PAUF*	GCACCACTGAAGACTACGACCAT	TGCAGGGTGACTTCCTGGGTATT
*ACTA2*	AAGACAGCTACGTGGGTG	GAGCAGGGTGGGATGCT
*VIM*	CGCCATCAACACCGAGTTCA	CCTTGAGCTGCTCGAGCT
*CDH1*	TCTGGATAGAGAACGCATTGC	GCTTGTTGTCATTCTGATCGGT
*TJP1*	CATCCACTCTGCTAATGCCT	GGAATGATCAGAAGGCTCTG
*GAPDH*	GGGGCTCTCCAGAACATCAT	GGTCAGGTCCACCACTGACA

## Data Availability

All data supporting the findings of this study are available upon request from the corresponding author.

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
