# Peer review of "TGF-β-Induced PAUF Plays a Pivotal Role in the Migration and Invasion of Human Pancreatic Ductal Adenocarcinoma Cell Line Panc-1"

_ijms, 2024, doi:10.3390/ijms252111420_

Round 1
Reviewer 1 Report
Comments and Suggestions for Authors
In general, the manuscript “TGF-β-induced PAUF Plays a Pivotal Role in Migration and Invasion of Human Pancreatic Ductal Adenocarcinoma Cell Line Panc-1” is well written and overall provides a good summary and explanation of the research problem and the results.
I find this research very interesting and with high potential to translate these findings to advance the therapeutic and diagnosis tools of pancreatic cancer further.
Specially, I think you created a very illustrative diagram showing the potential mechanism on action and summarizing your findings on the regulatory roles of PAUF.
Your conclusion is well pointed and nicely wraps up your research as well as proposing a novel and potential therapeutic target to inhibit pancreatic cancer progression.
Your methods and results have a very complete and concise data set taking different approaches to validate your hypothesis, by knocking in, knocking down and inhibiting the TGFB canonical and non-canonical pathways to highlight the effect of PAUF in regulation of invasion and migration.
My Overall recommendation is to Accept in present form
Author Response
Reviewer 1 :
In general, the manuscript “TGF-β-induced PAUF Plays a Pivotal Role in Migration and Invasion of Human Pancreatic Ductal Adenocarcinoma Cell Line Panc-1” is well written and overall provides a good summary and explanation of the research problem and the results.
I find this research very interesting and with high potential to translate these findings to advance the therapeutic and diagnosis tools of pancreatic cancer further.
Specially, I think you created a very illustrative diagram showing the potential mechanism on action and summarizing your findings on the regulatory roles of PAUF.
Your conclusion is well pointed and nicely wraps up your research as well as proposing a novel and potential therapeutic target to inhibit pancreatic cancer progression.
Your methods and results have a very complete and concise data set taking different approaches to validate your hypothesis, by knocking in, knocking down and inhibiting the TGFB canonical and non-canonical pathways to highlight the effect of PAUF in regulation of invasion and migration.
My Overall recommendation is to Accept in present form.
Response to comments :
We appreciate the Reviewer 1 for the positive comments.
Reviewer 2 Report
Comments and Suggestions for Authors
This manuscript investigated TGF-β/Smad signaling-induced PAUF expression in promoting pancreatic cancer cell EMT. The background information is well-introduced, and the methods are well-described. The results are presented in detail, and the conclusions are supported by the data. However, I have a few minor concerns.
Figures 1C and D show that PAUF mRNA and protein expression increased after TGF-β treatment. The authors demonstrated that the Smad-binding element interacts with the PAUF promoter to enhance its mRNA expression. However, does the increase in PAUF protein levels result solely from the rise in mRNA levels, or could TGF-β also influence the post-translational modification of PAUF?
Have the authors considered why TGF-β increases PAUF expression in Panc-1 cells but not in other cell lines? The conclusion may have limited significance if it only applies to a few pancreatic tumor cell lines. Can TGF-β increase PAUF expression in primary pancreatic tumor cells from clinical samples?
Author Response
Reviewer 2 :
This manuscript investigated TGF-β/Smad signaling-induced PAUF expression in promoting pancreatic cancer cell EMT. The background information is well-introduced, and the methods are well-described. The results are presented in detail, and the conclusions are supported by the data. However, I have a few minor concerns.
Reviewer comments #1:
Figures 1C and D show that PAUF mRNA and protein expression increased after TGF-β treatment. The authors demonstrated that the Smad-binding element interacts with the PAUF promoter to enhance its mRNA expression. However, does the increase in PAUF protein levels result solely from the rise in mRNA levels, or could TGF-β also influence the post-translational modification of PAUF?
Response to comments :
We appreciate the reviewer’s careful comments. In our study, we have confirmed that both PAUF mRNA and protein expression are increased through the activation of Smad by TGF-β. To confirm the correlation between the increase in Smad-mediated PAUF mRNA expression and protein expression, we performed loss-of-function experiments using a TGF-β receptor I kinase inhibitor (SB-431532) and Smads-targeting siRNA. As a result, we confirmed that both PAUF mRNA and protein expression were significantly suppressed by inhibiting TGF-β/Smad signaling. Based on these findings, we believe that the increase in PAUF protein expression occurs through the increase in mRNA expression, in accordance with the ‘central dogma theory’, rather than being influenced by post-translational modifications. Once again, we sincerely appreciate the reviewer for their valuable comments.
Reviewer comments #2:
Have the authors considered why TGF-β increases PAUF expression in Panc-1 cells but not in other cell lines? The conclusion may have limited significance if it only applies to a few pancreatic tumor cell lines. Can TGF-β increase PAUF expression in primary pancreatic tumor cells from clinical samples?
Response to comments :
Thanks for kind comment. In the results of Figure 2A, we confirmed that the five pancreatic ductal adenocarcinoma (PDAC) cell lines exhibited different responses to the TGF-β/Smad signaling cascade when stimulated with TGF-β. Among these, we observed an increase in PAUF expression only in the Smad signaling-positive Panc-1 cell line treated with TGF-β. The five PDAC cell lines are only a few of the various PDAC cell lines known to date. Therefore, it is difficult to conclude that the significance of PAUF expression and function through the activation of the TGF-β/Smad signaling pathway is limited in PDAC cells. In a previous case study involving 249 PDAC patients, it was revealed that 43% lacked Smad4 expression, while the 45% exhibited Smad4-positive expression and function [1,2,3]. Although it is difficult to determine the significance of TGF-β/Smad-mediated PAUF expression in cases with Smad4 defects (43%), the expression and function of PAUF through TGF-β/Smad signaling activation are expected to be significantly relevant in cases of Smad4-positive PDAC (45%). Therefore, we plan to investigate the correlation of PAUF expression induced by TGF-β using primary PDAC cell lines isolated from clinical samples, as suggested by the reviewer. We appreciate your insightful guidance.
References
- Stefanoudakis, D.; Frountzas, M.; Schizas, D.; Michalopoulos, N.V.; Drakaki, A.; Toutouzas, K.G. Significance of TP53, CDKN2A, SMAD4 and KRAS in Pancreatic Cancer. Issues Mol. Biol. 2024, 46, 2827–2844.
- Tascilar, M.; Skinner, H.G.; Rosty, C.; Sohn, T.; Wilentz, R.E.; Offerhaus, G.J.; Adsay, V.; Abrams, R.A.; Cameron, J.L.; Kern, S.E.; et al. The SMAD4 protein and prognosis of pancreatic ductal adenocarcinoma. Cancer Res.2001, 7, 4115–4121.
McCarthy, A.J.; Chetty, R. Smad4/DPC4. J. Clin. Pathol. 2018, 71, 661–664.